# The Impact of Oxidative Stress on Blood-Retinal Barrier Physiology in Age-Related Macular Degeneration

**DOI:** 10.3390/cells10010064

**Published:** 2021-01-04

**Authors:** Annamaria Tisi, Marco Feligioni, Maurizio Passacantando, Marco Ciancaglini, Rita Maccarone

**Affiliations:** 1Department of Biotechnological and Applied Clinical Sciences, University of L’Aquila, 67100 L’Aquila, Italy; annamaria.tisi@graduate.univaq.it; 2European Brain Research Institute, 00161 Rome, Italy; m.feligioni@ebri.it; 3Department of Neuroreabilitation Sciences, Casa di Cura Policlinico, 20144 Milano, Italy; 4Department of Physical and Chemical Sciences, University of L’Aquila, 67100 L’Aquila, Italy; maurizio.passacantando@univaq.it; 5Department of Life, Health and Environmental Sciences, University of L’Aquila, 67100 L’Aquila, Italy; marco.ciancaglini@univaq.it

**Keywords:** blood retinal barrier, oxidative stress, physiology, age-related macular degeneration

## Abstract

The blood retinal barrier (BRB) is a fundamental eye component, whose function is to select the flow of molecules from the blood to the retina and vice-versa, and its integrity allows the maintenance of a finely regulated microenvironment. The outer BRB, composed by the choriocapillaris, the Bruch’s membrane, and the retinal pigment epithelium, undergoes structural and functional changes in age-related macular degeneration (AMD), the leading cause of blindness worldwide. BRB alterations lead to retinal dysfunction and neurodegeneration. Several risk factors have been associated with AMD onset in the past decades and oxidative stress is widely recognized as a key factor, even if the exact AMD pathophysiology has not been exactly elucidated yet. The present review describes the BRB physiology, the BRB changes occurring in AMD, the role of oxidative stress in AMD with a focus on the outer BRB structures. Moreover, we propose the use of cerium oxide nanoparticles as a new powerful anti-oxidant agent to combat AMD, based on the relevant existing data which demonstrated their beneficial effects in protecting the outer BRB in animal models of AMD.

## 1. The Blood Retinal Barrier

As part of the central nervous system (CNS), the retina is particularly susceptible to alterations of its microenvironment, which can cause irreversible damage to vision. In addition, the retina, and especially the macula, is characterized by a state of physiological oxidative stress due to an elevated metabolism and high oxygen consumption [1]. Hence, the maintenance of a correct and balanced microenvironment is fundamental in order to allow the health of the retinal cells, especially of those of neuronal origin. For this reason, as well as in the brain, the retina is finely insulated from the bloodstream by a barrier, named the blood retinal barrier (BRB). It can be distinguished as an “inner BRB” and “outer BRB”, which regulate the permeability of substances at the inner and outer retina site respectively (Figure 1). Both structures are characterized by the presence of tight junctions, which prevents the uncontrolled diffusion of substances from the blood to the retina and vice-versa. In this way, molecules can cross the BRB mainly through a trans-cellular mechanism [2].

### 1.1. Inner BRB

The inner BRB is localized at the level of the retinal vasculature, which originates from the central artery, and supplies the inner retina. It enters the retina through the optic nerve and progressively branches along its layers excluding the photoreceptors. In fact, the photoreceptors layer lacks blood vessels and this is a fundamental characteristic in order to allow proper vision and health of photosensitive neurons [3]. The retinal vascular bed extends toward the retina at three main levels, the deep, intermediate, and superficial plexuses, which correspond to: the outer plexiform layer (OPL); the inner plexiform layer (IPL); and the nerve fibre layer (NFL) respectively [4].

The endothelial cells of the retinal vasculature are surrounded by a thick basement membrane and are joined by tight junctions. Therefore, the spaces between the endothelial cells are sealed, forming a barrier between the blood and the retina, and transport across the inner BRB occurs mainly through a trans-cellular mechanism [5]. Tight junctions consist of the proteins zonula occludens (ZO) 1–3 (ZO-1, -2, -3), cingulin, 7H6 antigen, occluding, symplekin, cadherin-5, and claudins [6,7]. Moreover, the endothelial cells are covered by pericytes, astrocytes, and Müller cells, which contribute to the inner BRB function and integrity. Pericytes are phagocytic and contractile cells which regulate vascular tone, support the capillary structure, and secrete extracellular molecules such as fibronectin [8]. The ratio of pericytes to vascular endothelial cells in the retina is higher than in other tissues [9]. Müller cells and astrocytes are also closely associated with the retinal endothelial cells. Because of their localization, both astrocytes and Müller cells interact with the vessels of the superficial plexus, forming the limitans glia. Conversely, the intermediate and deep plexuses are surrounded solely by Müller cells, which span along the entire retina and make contact with all the vascular plexuses [10]. Both astrocytes and Müller cells are fundamental for the maintenance of the vascular structure and for the release of growth factors and other proteins which contribute to the BRB properties [11,12]. In addition, microglia cells have also been recently demonstrated to be important mediators of the inner BRB integrity, contributing significantly to retinal vasculature development [13].

Damage to endothelial cells and to the inner BRB is a key event in several eye diseases, such as diabetic retinopathy, retinopathy of prematurity, retinal vein occlusion, and uveitis [14,15]. Importantly, the loss of the barrier integrity is also associated with neovascularization because of the absence of the characteristic blood-barrier structure in the new vessels [16]. The breakdown of the inner BRB is inevitably followed by vascular leakage and macular edema, which affects visual function and retinal cell health [17,18].

### 1.2. Outer BRB

Three major components constitute the outer BRB: the choroid, the Bruch’s membrane (BM), and the retinal pigment epithelium (RPE) [2]. The choroid supplies the outer retina and, differently from the inner BRB, shows multiple fenestrations and does not constitute a barrier itself. The restriction of permeability of the outer BRB from the choroidal blood flow is therefore allowed by additional structures which lie at the interface between the photoreceptors layer and the choroid: the RPE and the BM [2]. The outer BRB is the site where age-related macular degeneration (AMD) originates, and the understanding of the physiological and pathological processes occurring in the outer BRB are of crucial importance in order to develop effective therapies.

#### 1.2.1. The Choriocapillaris (CC)

The choroid shows a very high blood flow, greater than other tissues in the body, which is useful to meet the elevated metabolic demand from the retina [19]. It is located between the sclera and the BM and it is composed of several layers, from outer to inner: the suprachoroid, two vascular layers (large vessels and medium vessels), and the choriocapillaris (CC) [19]. The CC lies at the chorio-retinal interface and is responsible for the transport of nutrients to the photoreceptors and RPE cells, as well as for the removal of waste material. The CC vessels form a highly dense vascular network and show a large diameter and multiple fenestrations, which make them highly permeable to proteins and several macromolecules [20].

#### 1.2.2. The Bruch’s Membrane (BM)

The Bruch’s membrane (BM) is located between the choriocapillaris and the RPE, and it is a fibrous membrane composed of five layers, from outer to inner: the basement membrane of the choriocapillaris, the outer collagenous layer (OCL), the central elastic layer (EL), the inner collagenous layer (ICL), and the basement membrane of the RPE [19]. The RPE cells secrete the molecular components of the BM, whose major component is collagen. In particular, the basement membrane of the choriocapillaris is composed mainly of collagen IV, V, and VI together with laminin and heparin sulphate. The OCL and ICL show a similar composition, mainly characterized by collagen I, III, and V. The EL, as is evident from its denomination, is made up primarily of elastin fibres. The basement membrane of the RPE is composed of collagen IV, laminin, fibronectin, heparin sulphate, and chondroitin sulphate [21]. As part of the outer BRB, BM acts as a size-selective barrier, blocking the diffusion of molecules of high molecular size, which can eventually cross the BM through a passive diffusion mechanism. Moreover, the metabolic waste material deriving from photoreceptors and RPE cells crosses the BM to reach the choroid in order to be removed [22]. BM also prevents uncontrolled cell migration [23] and provides an attachment site for RPE cells through the basement membrane of the RPE [24]. In addition, BM plays an important role in the mechanical withstanding of physical forces and stress [25,26].

#### 1.2.3. The Retinal Pigment Epithelium (RPE)

The RPE is present between the BM and the photoreceptors [27] and is composed of post-mitotic pigmented polarized epithelial cells displaying an hexagonal morphology, which constitute a finely structured monolayer and play a pivotal role in the outer BRB function [28]. This is enabled by extensive zonulae occludens junctions, very similar to those present between the endothelial cells of the retinal vasculature, at the apical surface of RPE cells [29]. Transport across the RPE can occur through a para-cellular mechanism that is limited due to the semi-selective properties of the tight junctions, through an active transport which occurs against the electro-chemical gradient, and a facilitated diffusion which occurs following the electro-chemical gradient through selective transporters [30]. In addition to the barrier function, RPE cells also fulfil other important activities. Due to their multiple roles, RPE cells display several mitochondria, smooth- and rough-surfaced endoplasmic reticulum, and free ribosomes. The basal surface of RPE cells is characterized by numerous villi which penetrate between the photoreceptors’ outer segments (POS) [31]. This allows the shedding of POS, which is fundamental to allow their renewal and, in turn, proper vision [32]. RPE cells are also directly involved in the visual cycle thanks to the expression of peculiar enzymes and transporters, such as the retinal pigment epithelial 65 enzyme (RPE65) [33] and the ATP binding cassette subfamily A member 4 transporter (ABCA4) [34]. Moreover, the RPE cells play an important secretion activity releasing important growth factors, such as the pigment-derived epithelial factor (PEDF) [35].

## 2. Age-Related Macular Degeneration

Age-related macular degeneration (AMD) is the leading cause of blindness worldwide, accounting for about 7–8% of all blindness in the world, and its incidence is expected to increase in years to come [36]. AMD is a neurodegenerative disease of the retina, which affects primarily people older than 60 years, leading to irreversible central vision loss [37]. AMD occurs as a result of photoreceptor/RPE/BM/CC complex alterations, which culminates in BRB breakdown and retinal neurodegeneration [38]. Retinal degeneration starts in the macula, the retinal region responsible for visual acuity, and expands in size over time [37]. The exact pathogenesis of AMD has not been fully understood. Nonetheless, it has been well established that AMD can be considered a multifactorial disease. Aging, cigarette smoke, high fat diet, light exposure, alcohol consumption, and specific genetic polymorphysms (such as pigment-derived epithelial factor (PEDF)) are considered the main risk factors for AMD [39,40,41,42,43,44]. All these events share oxidative stress as a common feature that can be considered the driving force of all the risk factors [45].

### 2.1. AMD Classification

Several AMD classification systems exist [46,47]. In this review we refer to the conventional classification which distinguishes two major forms: the wet (exudative) and the dry (atrophic) AMD forms [48]. Depending on the clinical signs, AMD can be classified in an early or late (either dry or wet) pathology stage. Early AMD is commonly asymptomatic, while morphological signs are already detectable by colour fundus photograph. In particular, it is characterized by the accumulation of yellowish material in the macular subretinal space, known as drusen, which can increase in size over time [37]. The progression of the pathology and the increasing accumulation of waste material and toxic metabolites lead to advanced AMD, namely wet and dry AMD. The two AMD advanced forms are not exclusive and can occur concomitantly in the same patient. In fact, dry AMD can eventually progress to wet AMD, while wet AMD is often followed by retinal atrophy [48,49]. Dry AMD is the most common form, accounting for about 90% of all AMD patients [36]. It progresses slowly, and it is characterized by progressive outer BRB atrophy followed by death of the photoreceptors. To date, effective treatments to counteract dry AMD progression are not still available. Wet AMD is less frequent, accounting for about 10% of all AMD patients. It progresses faster than the dry AMD form and it is characterized by choroidal neovascularization (CNV), leading to proliferation of blood vessels into the photoreceptors’ layer, edema, haemorrhages, and cell death. Wet AMD is associated with increased vascular endothelial growth factor (VEGF) levels, which have been recognized to be the driving force of CNV. Hence, intravitreal injections of anti-VEGF monoclonal antibodies have shown a great degree of efficacy to prevent the progression of the pathology [50]. Nonetheless, several side effects are associated with anti-VEGF therapies, the first of which is the necessity of repeated and frequent intravitreal injections.

### 2.2. The Anti-Oxidant Machinery of the Retina and Its Implications in AMD

The retina is characterized by a physiological condition of oxidative stress due to the high retinal metabolism. Reactive oxygen species (ROS) are produced primarily in mitochondria and play important roles in physiological cell signalling [51], such as autophagy [52] and inflammation [53]. Nonetheless, it is important to maintain a balanced environment, in order to avoid retinal damage by excessive ROS release. Therefore, the retina puts several mechanisms in place to counteract the accumulation of ROS in order to self-protect. The retina has developed an antioxidant machinery consisting of specific enzymes (such as the cytochrome P450 mono-oxygenase system, superoxide dismutases (SOD), and catalases) and small molecular anti-oxidants (such as thiol, glutathione, and thioredoxin) [54]. Several transcription factors are involved in the regulation of this anti-oxidant machinery. Nuclear factor erythroid-2 related factor 2 (Nrf2), a basic leucine zipper transcription factor, is the most important regulator of the transcriptional program which coordinates the defences of the retina from oxidative stress [54]. The endogenous anti-oxidant defences are particularly important in the protection of RPE cells, which are exposed to high levels of oxidative stress due to their phagocytosis activity, which is associated with H_2_O_2_ production [55]. Recently, multiple evidences have demonstrated that the autophagy pathway is activated by oxidative stress and acts by removing defective cellular components, such as mitochondria damaged by ROS [56]. Impairment of autophagy is an important event in AMD [57], and initiates a vicious cycle, which culminates in caspase-mediated apoptosis [58,59,60]. Likewise, with aging, the physiological anti-oxidant defences decrease leading to the accumulation of toxic metabolites and free radicals, which in turn induce further oxidative stress [61]. On the other hand, it has been proposed that excessive stimulation of the antioxidant system could determine a Nfr2-mediated inflammatory condition, which can induce RPE cell death, accumulation of cellular debris, and occurrence of drusen [54].

### 2.3. Oxidative Stress and AMD

Oxidative stress, reactive oxygen species (ROS), and lipid peroxidation have a strong relationship with all the risk factors associated with AMD. Aging is considered the basic factor predisposing a person to AMD, and with increasing life expectancy in developed countries, AMD incidence continues to augment. This, in concomitance with other environmental and genetic factors, increases oxidative stress and the probability of developing AMD. In particular, cigarette smoke, the most important environmental risk factor for AMD, is a well-established source of oxidative stress and toxic material for all tissues. Cigarette smoke also induces RPE alterations and affects the expression level of specific proteins and growth factors [62,63]. A high fat diet, together with the reduction of the cholesterol and lipid elimination mechanisms due to aging, contributes to the lipid accumulation and, consequently, to increased lipid oxidation [64]. The excess of light exposure also represents an important risk factor for AMD [65,66]. Indeed, a widely accepted animal model of AMD consists of light-induced retinal degeneration, in which oxidative stress is a major player in the induction of the degeneration [67,68]. Finally, thanks to the outstanding advances in sequencing technologies and through genome-wide association studies (GWAS), significant improvements have been made in understanding the genetics underlying AMD. By 2017, 52 common and rare polymorphisms at 34 genetic loci had been identified to be independently associated with AMD, explaining over 50% of heritability [69]. In particular, single nucleotide polymorphisms (SNPs) which induce oxidative stress and inflammation have been identified as the major genetic variant involved in AMD development. The most frequent genetic variants that predispose a person to AMD concern the Complement Factor H (CFH) and Age-Related Maculopathy Susceptibility 2 (ARMS2) genes [69]. The gene product of ARMS2 has not yet been identified. Conversely, the gene product of CFH is well recognized and the protective role of CFH on the retina against oxidative stress has been extensively studied. The CFH CCP7 binding domain is mandatory in order to allow RPE protection from 4-HNE(4-hydroxy-2-nonenal)-induced cell death [70]. In addition to its ability to regulate the alternate complement pathway, the SCR7 domain is responsible for the CFH’s ability to bind lipid peroxidation products, such as malondialdehyde (MDA), released by the photoreceptors as a consequence of photo-oxidative stress [71]. The binding of CFH oxidized phospholipids is important in order to prevent the activation of oxidative stress-induced inflammation. Alterations in the CFH structure subsequently inhibit this important mechanism and participate in the chronic inflammation characteristic of AMD patients [72]. Moreover, the expression of CFH progressively decreases with age, driving oxidative stress exacerbation and promoting AMD [73]. Based on the relevant role that the oxidative stress plays in AMD pathogenesis, the supplementation of natural anti-oxidants (vitamins, ω-3 (n–3) fatty acids, carotenoids, etc.) represents a gold standard intervention for AMD patients. Indeed, the Age-Related Eye Disease Study (AREDS) anti-oxidant dietary supplementation has been associated with beneficial effects in AMD patients by reducing the risk of AMD progression to the advanced forms [74,75].

It can be summarized that the oxidative stress burden is a consequence of the exposure to genetic and environmental risk factors, and subsequently drives AMD development.

## 3. BRB Alterations in AMD

AMD is characterized by alterations of the CC/RPE/BM complex which culminates in the outer BRB breakdown and degeneration of the downstream photoreceptors in the macula. Each of the components of the outer BRB undergo peculiar structural and subsequently functional alterations, which are highly influenced by oxidative stress burden. Here, we report the main BRB changes in AMD and highlight the role of oxidative stress in driving their onset (summarized in Figure 2).

### 3.1. The Choriocapillaris Shows Different Characteristics in Wet and Dry AMD

At the early AMD stages some choroidal alterations already occur, including reduced blood flow and volume in the CC [76]. These CC alterations have been associated with increased drusen accumulation in the sub-RPE space [77]. In late AMD the CC is differently involved in AMD pathogenesis depending on the AMD type, namely wet or dry (for more details about the vascular contribution to AMD see Lipecz et al., 2019 [78]).

#### 3.1.1. The CC Thins in Dry AMD

Dry AMD is characterized by a high degree of choroidal/CC thinning [79,80], which inevitably affects the retrieval of nutrients as well as the homeostasis of the retinal environment. The choroid itself progressively thins with age and shows a reduced CC vascular density [81]. Intriguingly, in AMD patients, the area of vascular loss was shown to be more extended than that of RPE atrophy, preceding retinal degeneration [82,83]. This important evidence suggests a major role played by the choroid in triggering the progression of atrophic AMD, perhaps inducing RPE cell death. On the other hand, RPE alterations are conventionally considered the starting point of AMD [84]. There are already controversial data about this topic and more studies are needed to investigate the sequence of events in the outer BRB of patients suffering from atrophic AMD. Other CC alterations observed in late AMD include altered pericytes distribution and reduced endothelial fenestrations [85].

#### 3.1.2. Wet AMD Is Characterized by Choroidal Neovascularization (CNV)

Wet AMD shows a different scenario at the choroidal level. Indeed, wet AMD is characterized by choroidal neovascularization (CNV), which clearly highlights the substantial difference from atrophic AMD. In wet AMD increased levels of VEGF induce abnormal neoangiogenesis in the CC [86]. There are three subtypes of CNV in wet AMD and they are classified according to the site of the suspected invasion of the retina. Type 1 neovascularization arises when choroidal neovascularization occurs below the retinal pigment epithelium [87]. Type 2 neovascularization refers to choroidal neovascularization infiltrating into the photoreceptor layer and corresponds to classic choroidal neovascularization [87]. Type 3 neovascularization, also known as retinal angiomatous proliferation (RAP), occurs when retinal circulation is involved, with an anastomosis between the choroidal and retinal vessels [88]. The CNV lesions are often surrounded by areas of choriocapillaris nonperfusion, also known as a “dark halo” [89], suggesting that CNV could be the consequence of ischemia. In recent years, the development of anti-VEGF drugs has revolutionized the management of wet AMD, leading to efficient prevention and regression of CNV in patients [50]. However, some data indicate that repeated and continuous administrations of anti-VEGF drugs could drive the degeneration of the choroid and the loss of endothelial cells as observed in atrophic AMD [90,91,92]. 

#### 3.1.3. The Relationship between Oxidative Stress and CC Alterations

Several experimental data highlight a direct relationship between oxidative stress and choroidal alterations in AMD [93,94]. ROS induce the overexpression of VEGF in several retinal cell types, including RPE [95,96] and endothelial cells [97]. Moreover, in vivo and in vitro studies demonstrated that the suppression of ROS is associated with decreased VEGF expression [98,99,100,101]. On the other hand the oxidative stress is also associated with decreased choroidal thickness [102]. Nrf2 alterations have a major role in oxidative stress-induced CC changes. For instance, Nrf2-Knock out mice show CC abnormalities [103]. Most notably, clinical trials have demonstrated the beneficial effects of anti-oxidant supplementation for AMD patients, although anti-VEGF intravitreal injections represent the gold standard therapy for wet form AMD. Dietary anti-oxidants are conventional interventions to delay AMD progression in patients suffering from dry AMD [104].

### 3.2. Bruch’s Membrane Alterations in AMD

BM undergoes several changes that contribute to BRB dysfunction and breakdown in AMD patients [105]. The best-known event occurring in BM concerns the accumulation of waste material in the collagenous areas or in the intercapillary pillars, which cannot be eliminated by the CC as in physiological condition. Moreover, BM shows structural changes due to increased thickness and calcification. Taken together, all these factors lead to reduced permeability and elasticity of BM. As a consequence, BRB function is altered. Molecules cannot cross the BM properly, waste material cannot reach the CC to be eliminated, and there is a lack of effective gas and nutrient supply. On one hand, this unpleasant scenario triggers the progression to atrophic AMD due to the unavoidable damage of the retina. On the other hand, BM alterations can also trigger the progression to wet AMD due to hypoxia, which is followed by the induction of pro-angiogenic events [106]. In this context, a major role is played by VEGF, as described above, but also by matrix metallopeptidases (MMPs) which allow the destruction of BM extracellular matrix (ECM) and subsequent pathological CNV [21]. An important role is played by inflammation and especially by macrophages and other immune-related cells. They actively release pro-angiogenic factors and MMPs, and contribute to AMD progression, although their recruitment initially aims at resolving and protecting the eye from damaging events. As a consequence, chronic inflammation takes part in AMD pathogenesis [107].

#### 3.2.1. Bruch’s Membrane Deposits

The identity of BM deposits is heterogeneous and, although several advances have been made in this research area, the composition of BM deposits has not yet been exactly determined. The main BM deposits already established are drusen and basal deposits [108]. Drusen, as already mentioned in the previous paragraphs, are easily detectable by fundus examination due to their yellowish emission and they are supposed to be material undigested by the RPE and are constituted by lipids and more than 129 proteins. Amyloid-β has also been recently identified as an additional component of drusen. Many of the biomolecules identified in drusen have oxidative-modifications, supporting the importance of oxidative stress in AMD pathogenesis. Drusen can be classified as “hard” and “soft” depending on their morphology. Hard drusen are small (<63 µm), well defined deposits, that are often found in aged eyes regardless of AMD manifestation, and they generally localize in the intercapillary pillars. Conversely, soft drusen are larger than hard drusen (>125 µm) and show undefined edges. They usually aggregate forming bigger deposits and are associated with a worse prognosis of AMD development [37]. Basal deposits can be distinguished as basal linear deposits (BLinD) and basal laminar deposits (BLamD). BLinD localize between the RPE basement membrane and the ICL of the BM, and are composed primarily of membranous debris [109]. Depending on their thickness BLinD are specifically associated with the severity of AMD [109]. Conversely, BLamD localize between the basement membrane of the RPE and its plasma membrane, and constitute a thin layer of basement membrane proteins and long-spaced collagen fibres [110,111]. Other BM deposits include lipids, such as esterified cholesterol and lipoproteins. Together with drusen and basal deposits, lipid deposits ultimately form a “lipid wall” [112]. Waste material is also composed of proteins. Notably, about 60% of the accumulated proteins involve the activation of the immune system [108]. Other accumulated components include advanced glycation end products (AGEs) (glycosylated and oxidized fats and proteins), which accumulate on collagen fibres of BM [113], iron, and zinc. While it has been well established that iron accumulation contributes to several neurodegenerative diseases through the induction of oxidative stress, the effects of zinc accumulation have been not fully understood. Nonetheless, zinc is supposed to be involved in AMD due to its presence in sub-RPE and BM deposits [21].

#### 3.2.2. Bruch’s Membrane Thickness

In addition to the accumulation of waste material, BM also undergoes other important structural changes. Increased thickness is a major event observed in AMD eyes [114,115]. This is in part due to the accumulation of waste material. However, progressive BM thickness is observed with ageing, and it doubles throughout life [21]. BM thickness is also favoured by collagen cross-linking, which increases the density of collagen fibres [21], and by BM calcification, which is due to calcium and magnesium deposition on drusen [116,117].

### 3.3. The Oxidative Stress Plays a Central Role in AMD-Related RPE Alterations

The involvement of RPE alterations in AMD pathogenesis has been extensively studied in the past decades. To date, the RPE is considered to be the main protagonist of both dry and wet AMD, and several therapeutic strategies aim at protecting the RPE or eventually replacing it in late stages when it is almost degenerated. RPE cells undergo functional and morphological changes starting from early AMD, culminating in extensive cell death in the most advanced stages.

#### 3.3.1. RPE Changes in Early AMD

In the progression from early to late AMD, oxidative stress triggers several RPE alterations. It is widely accepted that oxidative stress drives an “energetic crisis” in RPE cells, which is strongly associated with mitochondrial damage [84,118]. As mentioned above, due to the multiple functions carried out by the RPE, it shows a high metabolic demand. The concomitance of multiple risk factors culminates in a failure of RPE performance and initiates the drop towards AMD. Morphologically, RPE cells start to accumulate undigested material, especially deriving from POS phagocytosis and damaged intracellular organelles. This leads to the formation of autofluorescent material, known as lipofuscin, which is clearly detectable by fundus autofluorescence imaging [119]. Lipofuscin, whose main constituent is N-retinyl-N-retinylidene ethanolamine (A2E) [120], is also composed of oxidized cross-linked proteins, lipids, and other waste material [121]. Lipofuscin induces photosensitization of RPE cells which culminates in oxidative stress burden and further cellular damage. On the other hand, a strong relationship has been identified between lipofuscin and cellular senescence [122], autophagy [56,123], and pro-angiogenic signalling [124]. Autophagy is physiologically relevant for RPE cells mediating the removal of damaged organelles and is involved in the process of POS phagocytosis, through a non-canonical autophagy mechanism called LAP (LC3-associated phagocytosis) [125]. Under stress conditions, the autophagy flux is altered. Specifically, it has been demonstrated that autophagy activation acts as a protective mechanism in AMD through inducing the removal of waste material [56]. On the other hand, other studies indicate that sustained activation of autophagy is linked with cell death [126]. Moreover, it has been demonstrated that oxidative stress is a major inductor of autophagy [52]. Therefore, further studies are needed in order to elucidate the role of autophagy in AMD pathogenesis and eventually develop targeted therapies.

#### 3.3.2. RPE Changes in Late AMD

With the progression of the pathology, RPE cells undergo other morphological and functional alterations. Recently, the epithelial-mesenchymal transition (EMT) has been identified as a major event underlying BRB breakdown in AMD. EMT is a process of RPE de-differentiation, in which RPE cells lose their peculiar RPE features, including the structural properties which enable the integrity of the BRB [127] (Figure 3). The destruction of tight junctions is inevitably followed by unbalanced gas and nutrient exchange, as well as uncontrolled diffusion of molecules and cells (either inflammatory and endothelial cells). Some data indicate a relationship between autophagy alterations, cell senescence, and EMT in AMD. Likewise, oxidative stress has been identified as another event triggering EMT, autophagy, and cell senescence [58,128,129,130]. Moreover, exacerbated oxidative stress drives functional RPE alterations, such as the oxidation of proteins and lipids, and induces mitochondrial DNA damage. This leads to the inability to digest metabolites or aged organelles, which are released at the basal surface of the RPE to the BM. As a consequence, this waste material cannot properly cross the BM and accumulates forming the different BM deposits [21]. In the final stage, AMD is characterized by extensive RPE and photoreceptors cell loss, a condition known as “geographic atrophy” [131].

### 3.4. Contribution of BRB Alterations to Impaired Retinal Function in AMD

#### 3.4.1. Application of Electroretinography (ERG) to Investigate Retinal Dysfunction Due to BRB Alterations in AMD Patients

Given the strong relationship between the outer BRB and the retinal homeostasis, any BRB alterations can affect retinal function even before evident morphological changes. Retinal function can be investigated through several techniques and protocols of electroretinography (ERG) in AMD patients. ERG allows the recording of the electrical response of the retina to different visual stimuli. It is important to note that this technique requires age-matched controls for clinical AMD studies since retinal function is already worse in the elderly [132,133]. This is further corroborated by the evidence that choroidal thickness progressively decreases with age as is also described in Section 3.1.1, and a higher choroidal thickness has been positively associated with better visual acuity [134]. Full-field (or flash) ERG (ffERG) is conducted by stimulating the retina through flashes of light. From ffERG several waves are obtained, including the a-wave (a negative deflection corresponding to photoreceptors activity), the b-wave (a positive wave corresponding to the inner retina activity) [135], and the c-wave (a late-onset positive wave corresponding to RPE function) [136]. ffERG elicits an overall response from all retinal regions. Hence, ffERG does not allow the distinguishing of limited or little retinal alterations and has a poor sensibility. Nonetheless, ffERG recordings show decreased amplitudes and increased implicit times of a- and b-waves and it is therefore a useful tool to detect retinal functional alterations in AMD patients [137]. Conversely, focal ERG (fERG) stimulates the response exclusively from the fovea and it is therefore specific for macular diseases [137]. Multifocal ERG (mfERG), instead, allows simultaneous stimulation of multiple retinal regions, whose electrical response can be dissected. To date mfERG is considered to be the best ERG technique to study retinal function in AMD thanks to its high sensibility [137]. As widely described in this review, the BRB is a fundamental supporting structure for the neuroretina. Therefore, any BRB alterations have repercussions on retinal function. At early AMD stages, when patients are still asymptomatic, retinal functional impairment can already be detected by ERG [134]. At this stage, only few morphological alterations are visible, such as drusen accumulation or increased fundus autofluorescence. However, several undetectable alterations are still ongoing in the outer BRB and will lead to symptoms in more advanced stages. Hence, the study of retinal function in AMD represents an interesting way to diagnose AMD. In late AMD, retinal function is irreversibly impaired and there is a condition of complete central vision loss due to massive outer BRB and neuroretinal degeneration. Many evidences indicate a direct relationship between ERG impairment and AMD, either wet [138] and dry [139]. Moreover, a worse retinal function correlates with increased drusen accumulation and increased macular thickness in patients, as observed through optical coherence tomography (OCT) [139]. Several studies imply the use of mfERG not only for AMD diagnosis, but also as a major tool to investigate the progression of the disease or to manage the outcome of surgical/pharmacological interventions [140,141,142].

#### 3.4.2. Studies on Animal Models of AMD Elucidate the Mechanisms Underlying Retinal Dysfunction Due to BRB Changes

Studies on animal models of AMD have allowed deeper understanding of the relationship between outer BRB damage and retinal dysfunction. In particular, it is possible to directly investigate RPE function through the analysis of the c-wave deriving from ffERG stimuli. For instance, the c-wave was affected in a model of AMD induced by mitochondrial oxidative stress, and was restored after antioxidant supplementation, supporting the importance of the oxidative stress in triggering BRB damage [143]. Sodium Iodate (NaIO_3_)-induced RPE damage also correlated with decreased c-wave amplitude, and protection of the RPE by cytokeratin 17 restored normal RPE function [144]. Despite the relevant information that the c-wave provides, it is not usually investigated in human subjects due to methodological problems and discomfort felt by patients. Additionally, a- and b-waves have been shown to be altered in animal models of AMD, as well as in humans, even before evident morphological changes. For instance, CFH^−/−^ mice displayed decreased a- and b-waves amplitudes, which were associated with abnormal mitochondria in the RPE by electron microscopy, indicating a metabolic dysfunction [145]. Injection of amyloid-β peptide in C57BL/6 mice induced a drastic ERG impairment (a-,b-, and c-waves) associated with RPE senescence, increased BM thickness, and the overexpression of inflammatory genes in the RPE/choroid [146]. Bruch’s membrane alterations and RPE destruction, together with impaired retinal function, were also found in C57BL/6 mice after the intravitreal injection of adeno-associated virus (AAV) expressing a rybozime which targets SOD2 [147]. Nrf2^−/−^ mice showed RPE alterations, including vacuolation, hyper-and hypo-pigmentation, and cell death. Moreover, age-dependent drusen and BlamD-like accumulation was found in the subretinal space of these mice, together with CNV. All these features correlated with impaired retinal function in the elderly, either a- and b-waves [103]. This data indicates once again that the reduction of the antioxidant defences impacts the BRB function and vision.

## 4. Nanoceria: A New Nanotechnology Approach to Combat AMD-Induced BRB Breakdown

Based on the absence of effective therapies for the treatment of dry AMD and on the several side effects deriving from anti-VEGF injections in wet AMD eyes, there is an increasing interest in developing new therapeutic strategies. In recent years important improvements have been made surrounding nanomedicine, which represents a promising research field due to the unmatched properties of nanoparticles. Several nano-based drugs have been developed and each of them show peculiar chemical and physical properties, which make them unique. Among these, cerium oxide nanoparticles (CeO_2_-NPs) have been extensively studied in the past decades as a therapy to prevent retinal degeneration thanks to their anti-oxidant properties [148]. In this paragraph we describe the characteristics of cerium oxide nanoparticles and their applications as a promising therapy to combat AMD, targeting the oxidative stress-induced outer BRB breakdown.

### 4.1. Cerium Oxide Nanoparticles Show Peculiar Anti-Oxidant Properties

Cerium oxide nanoparticles (CeO_2_-NPs) are synthesized from cerium, an earth element belonging to the series of lanthanides, according to a defined protocol previously published [149]. The synthesis protocol allows the obtaining of a non-stoichiometric compound (CeO_2−x_) that has the ability to switch from an oxidation state to another, due to the presence of oxygen vacancies on the nanoparticles’ surface. Indeed, cerium shows two oxidation states, Ce^3+^ and Ce^4+^, and this bidirectional transition allows the scavenging of free radicals. Thanks to this important feature, CeO_2_-NPs are pure anti-oxidant agents and show this important property without exhaustion in an “auto-regenerative” manner [150]. These peculiar chemical and physical features make CeO_2_-NPs a promising and exciting compound for biomedical applications. This is further favoured by the absence of toxicity in several tissues as demonstrated by multiple studies either in vitro and in vivo [151]. Nonetheless, some data indicate a certain degree of citotoxicity and therefore further study is needed to deepen the biocompatibility of CeO_2_-NPs [152]. Nevertheless, the retina represents an advantageous tissue for CeO_2_-NPs applications for several reasons. First of all, the presence of a BRB prevents the uncontrolled diffusion of the nanoparticles in the bloodstream after intravitreal injection. This is confirmed by several studies that demonstrated the persistence of CeO_2_-NPs in the retina for a long time (up to one year) [153,154]. Moreover, the retina is an immune-privileged tissue and is characterized by an immune suppressive microenvironment, which allows its protection from gliosis insults [155]. This condition prevents unwanted responses against CeO_2_-NP, which would cause retinal damage. On this basis, several studies demonstrated that CeO_2_-NPs are biocompatible and well tolerated by the retina, paving the way for future ocular applications [153,156].

### 4.2. Cerium Oxide Nanoparticles are a Promising Therapy for AMD and Preserve Retinal Function

CeO_2_-NPs have been tested in several animal models of eye diseases, such as AMD, retinitis pigmentosa (RP) [148], cataracts [157], and corneal neovascularization [158]. Multiple promising data have been produced especially about the efficacy of CeO_2_-NPs as a therapy to combat AMD, preventing several BRB alterations (Table 1). Chen J. and colleagues first demonstrated that intravitreal injections of CeO_2_-NPs in the light damage (LD) model of AMD prevent retinal degeneration and this is associated with reduced intracellular peroxides and preserved retinal function [159]. The LD model reproduces several AMD features, including RPE degeneration [160], lipofuscin accumulation [161], photoreceptors’ death [67], and vascular alterations [162], and it is therefore a suitable model to test new therapies. Further studies in the same animal model demonstrated that CeO_2_-NPs maintain their neuroprotective properties for a long time and also target microglia activation preventing chronic inflammation [149,154,163]. Moreover, flash electroretinogram recordings (ERG) showed that retinal function was maintained when the treatment was performed before light exposure [149,159], and recovered when CeO_2_-NPs were injected after LD [154]. CeO_2_-NPs were recently tested in a transgenic model of AMD lacking Nrf2 (Nrf2^−/−^ mice) and exposed to LD [164]. As described in Section 2.2, Nrf2 is an important regulator of the antioxidant self-defences, therefore Nrf2^−/−^ mice display an AMD-like phenotype in the elderly due to increased oxidative stress. If subjected to LD, retinal degeneration occurs faster in Nrf2^−/−^ mice [164]. The authors demonstrated that CeO_2_-NPs, coated with glycol chitosan in order to improve water solubility, were effective in protecting Nrf2^−/−^ mice undergoing LD [164].

#### 4.2.1. CeO_2_-NPs Prevent Choroidal Neovascularization

CeO_2_-NPs represent a promising therapy to prevent pathological neovascularization. Hence, a first study in Very-low-density lipoprotein receptor (Vldlr)^−/−^ mice, a suitable model for wet AMD because of the onset of intraretinal and subretinal neovascularization, demonstrated that CeO_2_-NPs prevent oxidative damage. As a consequence, CeO_2_-NPs inhibited neoangiogenesis and VEGF overexpression [167]. In addition, further studies in the same animal model demonstrated that CeO_2_-NPs’ neuroprotection was associated with decreased expression of pro-angiogenic and inflammatory genes [166]. More recently it has been demonstrated that CeO_2_-NPs prevent laser-induced choroidal neovascularization in vivo [168]. The authors showed that this important result was associated with reduced oxidative damage and VEGF down-regulation [168]. On this basis, CeO_2_-NPs could have the potential to prevent CC alterations in AMD by inhibiting pathological neovascular processes.

#### 4.2.2. CeO_2_-NPs Prevent RPE Alterations

Only recently our group showed for the first time that CeO_2_-NPs have the ability to prevent BRB breakdown through the protection of the RPE in the LD model of AMD [165]. Notably, CeO_2_-NPs prevented light-induced EMT and autophagy alterations in RPE cells [165]. This important evidence was associated with CeO_2_-NPs’ localization in the RPE cytoplasm [165,169]. Moreover, intravitreal injections of CeO_2_-NPs also reduced subretinal autofluorescent deposits in the same model. Based on the CeO_2_-NP localization and subsequent RPE protection, it is possible that those autofluorescent granules included lipofuscin [170]. Intriguingly, another in vitro study demonstrated that CeO_2_-NPs have the ability to protect RPE cells from iron-induced oxidative stress [171]. To date, effective treatments to prevent RPE alterations, as well as lipofuscin deposition, in AMD do not exist. On this basis, CeO_2_-NPs represent an exciting opportunity to develop an effective therapy to achieve this purpose.

## 5. Concluding Remarks and Future Perspectives

AMD is a complex multifactorial disease which affects the outer BRB physiology, leading to irreversible retinal damage and vision loss. Current knowledge of the pathophysiological processes underlying AMD highlights the major role of oxidative stress in triggering outer BRB degeneration, as widely demonstrated by several studies in vitro and in vivo. This is further corroborated by clinical trials on human subjects experiencing beneficial effects through anti-oxidant supplementation. In this context, the use of anti-oxidant nanotechnologies could represent a promising advancement for the management of AMD patients and further investigations should continue in this direction. The dynamics of events that arise between the CC, the BM, and the RPE include complex mechanisms, many of which are yet to be discovered. Therefore, further information on the specific mechanisms established in AMD eyes is required in order to develop more efficient therapies and prevent patients from complete blindness. For instance, the analysis of retinal function allows the obtaining of additional information on retinal health, even if clear signs of AMD are not present. This raises a paradoxical question that scientists in the field need to answer: what happens in blind eyes that we are not still able to see?

## Figures and Tables

**Figure 1 cells-10-00064-f001:**
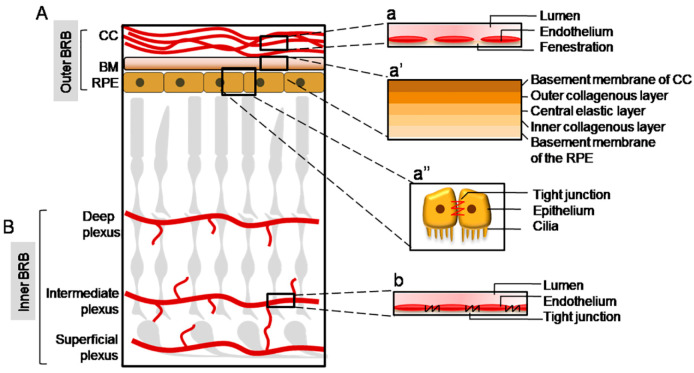
Schematic illustration of the blood-retinal barrier (BRB) structure. (**A**) The outer BRB insulates the outer retina from the bloodstream deriving from the choriocapillaris (CC), whose endothelium shows multiple fenestrations (a). The molecules reach the neuroretina through the Bruch’s Membrane (BM), composed of five layers, which forms a size-selective barrier (a’), and through the retinal pigment epithelium (RPE), characterized by tight junctions (a’’). (**B**) The inner BRB insulates the inner retina from the retinal vasculature, composed by a deep, intermediate, and superficial plexus; the endothelial cells of the retinal vessels form an effective barrier because of the presence of tight junctions among them (b).

**Figure 2 cells-10-00064-f002:**
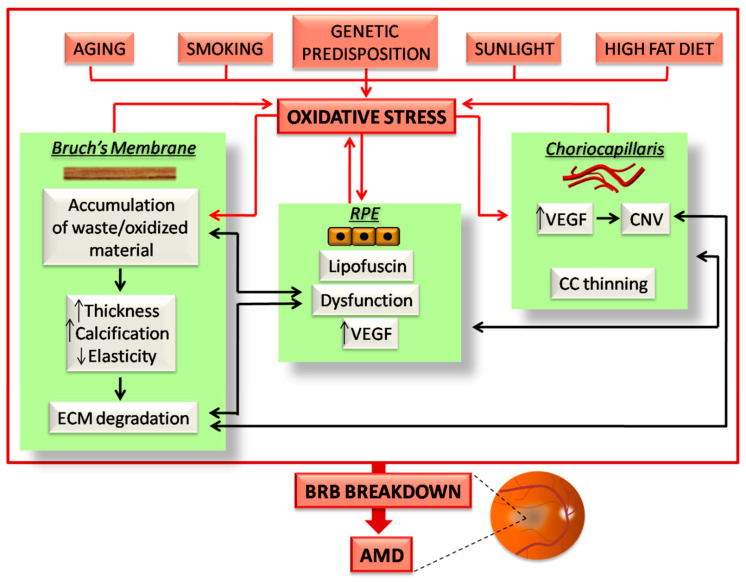
Schematic illustration of the events occurring in the outer BRB due to oxidative stress in age-related macular degeneration (AMD). All the risk factors for AMD induce oxidative stress burden, which affects the three components of the outer BRB. The effects of oxidative stress on the CC/BM/RPE complex culminates in a vicious cycle, which in turn induces further oxidative stress (red arrows). In addition, the induced damaging events of each BRB component inexorably affects the function and structure of the others (black arrows). All of these events highlight the finely balanced nature of the outer BRB, in which any alterations trigger a series of sequential events, which ultimately induce BRB breakdown and AMD. A detailed description about how oxidative stress affects the outer BRB in AMD is reported in the following paragraphs.

**Figure 3 cells-10-00064-f003:**
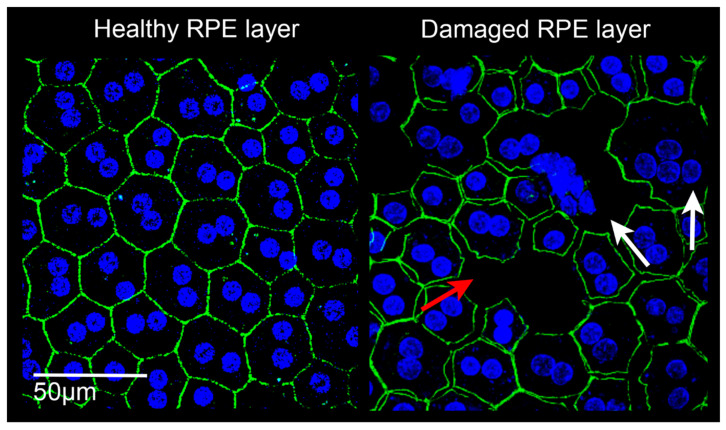
Confocal images of retinal pigment epithelium (RPE) in rats. **Left**: RPE from a healthy albino Sprague Dawley rat. The RPE is constituted by a homogeneous monolayer of joined epithelial cells displaying a hexagonal morphology. **Right**: RPE from a light damaged albino Sprague Dawley rat. The RPE structure appears clearly altered, cell junctions are lacking (red arrow), the onset of multinucleation suggests epithelial-mesenchymal transition (white arrows), and some cells have degenerated. Images were acquired by Leica TCS SP5 confocal microscope; 60× magnification. Green: phalloidin staining; blue: bisbenzimide nuclear dye; scale bar: 50 µm.

**Table 1 cells-10-00064-t001:** Summary of the protective effects of CeO_2_-NPs on BRB components (CC, BM, RPE) in animal models of AMD.

Animal Model	CC	BM	RPE	Ref.
Light damage(LD)	NA	Reduced accumulation of waste material	Prevented EMT	[165]
Nrf2^−/−^ mice undergoing LD	NA	Prevented BM thickening	Prevented lipofuscin accumulation	[164]
Vldlr^−/−^ mice	Inhibition of intraretinal and subretinal neovascularization	NA	NA	[166,167]
Laser-induced CNV	Inhibition of CNV	NA	NA	[168]

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
