# Peer review of "The Impact of Oxidative Stress on Blood-Retinal Barrier Physiology in Age-Related Macular Degeneration"

_cells, 2021, doi:10.3390/cells10010064_

Round 1

Reviewer 1 Report

My comments

1- Long paragraphs should be avoided as in page 4, 5

2- Add more clinical studies regarding the effect of oxidative stress on BRB

3-The authors focus more on outer BRB, more details on effects of oxidative stress on inner BRB in AMD should be added

Author Response

  • Long paragraphs should be avoided as in page 4, 5

Reply: We thank the reviewer for the useful comment. To limit the length of paragraph 2 we separated it in sub-paragraphs.

  • Add more clinical studies regarding the effect of oxidative stress on BRB

Reply: Studies on the effect of the oxidative stress on BRB are primarily based on basic research, while clinical studies on this topic mainly concerns the supplementation of anti-oxidants as a therapy to combat AMD progression. We added more details about it in paragraph 2.3 (lines 244-249).

3-The authors focus more on outer BRB, more details on effects of oxidative stress on inner BRB in AMD should be added

Reply: It is widely known that AMD is a pathology which targets the CC/BM/RPE complex, that is the outer BRB, while the inner BRB is not affected. On this basis the request of the reviewer cannot be met. 

Reviewer 2 Report

The review by Tisi and collaborators beautifully review the current literature on the role of oxidative stress on BRB and the impact of oxidative stress. The MS is very well written and clear. I would only suggest to add some information on ARMS2 function in paragraph 2.2. The authors, indeed, nicely describe the role of CFH, but no mention to the effects of ARMS2 polymorphism on retinal function is given.

Author Response

The review by Tisi and collaborators beautifully review the current literature on the role of oxidative stress on BRB and the impact of oxidative stress. The MS is very well written and clear. I would only suggest to add some information on ARMS2 function in paragraph 2.2. The authors, indeed, nicely describe the role of CFH, but no mention to the effects of ARMS2 polymorphism on retinal function is given.

Reply: The function of the protein product derived from ARMS2 gene has not been identified yet (paragraph 2.3.; lines 233-234).

Reviewer 3 Report

This is an excellent review article focusing on oxidative and BRB function in AMD pathogenesis. The authors even mentioned that cerium oxide nanoparticles as a potential therapeutic against AMD.

I have only one comment on cerium oxide nanoparticle as a potential therapeutic on AMD. There are still in debate on the safety of nanoparticles. It has been reported that there are toxicities of TiO2 (the most common used nanoparticles). How can the author confirm that there is no toxic effect of cerium oxide nanoparticle on photoreceptor cells? Moreover, there are many other anti-oxidative stress compounds (natural compounds) showed anti-oxidation and beneficial effects on AMD. The author should include the discussion on the natural compounds. Moreover, the authors should also describe the safety of cerium oxide nanoparticles against photoreceptor cells. 

Author Response

This is an excellent review article focusing on oxidative and BRB function in AMD pathogenesis. The authors even mentioned that cerium oxide nanoparticles as a potential therapeutic against AMD.

I have only one comment on cerium oxide nanoparticle as a potential therapeutic on AMD. There are still in debate on the safety of nanoparticles. It has been reported that there are toxicities of TiO2 (the most common used nanoparticles). How can the author confirm that there is no toxic effect of cerium oxide nanoparticle on photoreceptor cells? Moreover, there are many other anti-oxidative stress compounds (natural compounds) showed anti-oxidation and beneficial effects on AMD. The author should include the discussion on the natural compounds. Moreover, the authors should also describe the safety of cerium oxide nanoparticles against photoreceptor cells.

Reply: The safety of cerium oxide nanoparticles for the retina (and photoreceptor cells) was demonstrated in previous studies (PMID: 23536794; 27746672) that are mentioned in paragraph 4.1 line 515. We also added a mention about the effects of anti-oxidative stress compounds in AMD patients (paragraph 2.3.; lines244-249).